# The Response and Survival Mechanisms of *Staphylococcus aureus* under High Salinity Stress in Salted Foods

**DOI:** 10.3390/foods11101503

**Published:** 2022-05-22

**Authors:** Ying Feng, Tinghong Ming, Jun Zhou, Chenyang Lu, Rixin Wang, Xiurong Su

**Affiliations:** 1State Key Laboratory for Managing Biotic and Chemical Threats to the Quality and Safety of Agro-Products, Ningbo University, Ningbo 315211, China; fengying19830114@thnu.edu.cn (Y.F.); mingtinghong@nbu.edu.cn (T.M.); zhoujun1@nbu.edu.cn (J.Z.); luchenyang@nbu.edu.cn (C.L.); suxiurong_public@163.com (X.S.); 2College of Life Sciences, Tonghua Normal University, Tonghua 134000, China; 3School of Marine Sciences, Ningbo University, Ningbo 315211, China

**Keywords:** *Staphylococcus aureus*, salted foods, salt stress

## Abstract

*Staphylococcus aureus* (*S. aureus*) has a strong tolerance to high salt stress. It is a major reason as to why the contamination of *S. aureus* in salted food cannot be eradicated. To elucidate its response and survival mechanisms, changes in the morphology, biofilm formation, virulence, transcriptome, and metabolome of *S. aureus* were investigated. *IsaA* positively regulates and participates in the formation of biofilm. Virulence was downregulated to reduce the depletion of nonessential cellular functions. Inositol phosphate metabolism was downregulated to reduce the conversion of functional molecules. The MtsABC transport system was downregulated to reduce ion transport and signaling. Aminoacyl-tRNA biosynthesis was upregulated to improve cellular homeostasis. The betaine biosynthesis pathway was upregulated to protect the active structure of proteins and nucleic acids. Within a 10% NaCl concentration, the L-proline content was upregulated to increase osmotic stability. In addition, 20 hub genes were identified through an interaction analysis. The findings provide theoretical support for the prevention and control of salt-tolerant bacteria in salted foods.

## 1. Introduction

In traditional food processing systems, salting methods are often used to prevent food spoilage [1]. Previous studies demonstrated that an increased salt concentration in food can effectively reduce the water activity of food-borne pathogens, thereby inhibiting their growth [2]. Therefore, the water activity of salted foods is usually regulated by the addition of salt. This method can reduce the metabolic activity of microorganisms, so as to better control the spoilage rate of food [3]. *Staphylococcus aureus* (*S. aureus)* is considered to be the most important pathogen in salted food processing, and commonly causes food poisoning [4]. However, the health standards of various countries limit the detection of *S. aureus* in food, meaning that there are still many batches of food that exceed the prescribed limit [5]. *S. aureus* has a strong tolerance to high salt stress. It is major reason as to why the contamination of *S. aureus* in salted food cannot be eradicated [1]. Therefore, both the prevention and control of *S. aureus* present important research topics in food safety.

*S. aureus* is widely distributed and demonstrates strong adaptability to the environment [6]. It can survive in soil, air, food, and skin and has resistance to drying, heat, salt, alkali, and low temperatures [7]. In addition, when *S. aureus* is permitted to grow in foods, it produces a large number of toxic extracellular proteins, enterotoxins, and hemolysins, which are destructive to the intestinal tract and can seriously endanger human safety and health [8,9]. The ability of *S. aureus* to quickly adapt to environmental changes is also one of the reasons for its high pathogenicity [10].

*S. aureus* can adapt and survive in a salt environment. It was previously demonstrated that biofilm is an extracellular polymeric matrix secreted by certain microorganisms, such as *S. aureus*, and its production is well correlated with NaCl concentrations [11]. Recent research revealed that variations in NaCl concentrations can induce the increased production of biofilm in *S. aureus*, accompanied by the increased expression of genes related to biofilm, such as the *icaA* gene [12]. Furthermore, biofilm formation can provide *S. aureus* with a protective environment that prevents direct contact between salt and the bacterial cell membrane, allowing bacteria to maintain a growth state in adverse environments [13]. Additionally, numerous studies indicate that many small molecules referred to as compatible solutes, such as glutamine, proline and glycine betaine, can intracellularly accumulate in *S. aureus* due to increased biosynthesis or uptake, and decreased degradation modes, resulting in an improvement in osmotolerance [14,15].

*S. aureus* demonstrates a strong tolerance to high salt. Ming et al. conducted a preliminary study on the growth of *S. aureus* when subjected to high salinity stress based on proteomics [16]. However, the regulation mechanism of gene expression and the regulation of metabolites remain unclear. The adaptive changes of *S. aureus* after high salinity stress have not been systematically and comprehensively studied.

Generally, the salt concentration in salted food needs to be higher than 15% to inhibit bacterial growth. At present, the research on the mechanism of *S. aureus* salt tolerance is mainly limited to the low salt concentration range [12,15,17]. Therefore, in this study, we simulated salt stress with culture conditions of 0% NaCl, 10% NaCl, and 20% NaCl. Transcriptomic and metabolomic analyses were integrated to understand the responses and survival mechanisms of *S. aureus* under high salinity stress. This study presents genes, metabolites, and pathways that can be used as potential targets for preventing *S. aureus* contamination. The purpose of the study is to provide theoretical support for the prevention and control of salt-tolerant bacteria in salted food.

## 2. Materials and Methods

### 2.1. Strain and Culture Conditions

Commercial *S. aureus* ATCC 27217 was purchased from the Chinese Center of Industrial Culture Collection. The strain was incubated in a nutrient broth (NB; Guoyao, China) medium at 37 °C with shaking (140 rpm). For the experiment, equal volumes of the cells were inoculated in a 100 mL NB medium supplemented with 0% NaCl (control group), 10% NaCl, and 20% NaCl for 48 h. Three replicates were performed for each concentration culture. Bacterial liquids under different salt concentration stresses were used for subsequent experiments. Cell growth was monitored every 4 h. The optical density (OD) at 600 nm was used to monitor the growth of *S. aureus* in different conditions. 

### 2.2. Scanning Electron Microscopy (SEM)

The harvested cells (12,000 rpm, 2 min) were fixed with 2.5% glutaraldehyde at 4 °C for 2 h and washed three times with PBS buffer. Then, different concentrations of ethanol (30%, 50%, 70%, 90%, 100%) were used for the gradient dehydration of cells, and acetone was dehydrated twice. Finally, the suspension of acetone and cells was dropped on a coverslip to dry and fix. Gold coating was performed with a Zhongke model BS2 surface treatment machine. Images were captured with a Hitachi model S3000N scanning electron microscope. A total of 9 images of the 3 samples were obtained with SEM. 

### 2.3. Biofilm Analysis

The volume of bacterial liquid under different salt concentration stresses was adjusted to obtain a final cell count of 5 × 10^6^ CFU/mL in the NB medium. Then, the 3M^TM^ Petrifilm^TM^ Rapid *S. aureus* Count Plate method was used for cell counting. The bacterial solution (1 mL) was collected by centrifugation at 12,000 rpm for 2 min. First, the cells were washed three times with PBS buffer and fixed in 2.5% glutaraldehyde at 4 °C for 1.5 h. Then, cells were washed again and *S. aureus* was stained with FITC-labeled Concanavalin A (FITC-ConA; Sigma, St. Louis, MO, USA) for 30 min and Propidium Iodide (PI; Sigma, St. Louis, MO, USA) for 15 min at 4 °C. Finally, coverslips were mounted with 20 μL of the Antifade Mounting Medium (P0126, Beyotime, Shanghai, China). Biofilm formation was observed under confocal laser scanning microscopy (CLSM) immediately after tableting.

FITC-ConA and PI were used for fluorescence staining in the dark. Extracellular polymeric substances (EPSs) bind with FITC-ConA to emit green fluorescence (λex = 488 nm and λem = 520 nm), and damaged or dead cell DNA bind with PI to emit red light (λex = 543 nm and λem = 572 nm). Images were captured with the CLSM Zeiss model LSM880 microscope. 

The Easy3D function of the IMARIS 9.0 software package was used for the 3D reconstruction of the biofilm structure. The thicknesses of the biofilms were determined using the confocal stack images. Z stack represents the thickness of biofilm. Biofilm thickness measurements were performed with Zen black 2.1 software. Three replicates were performed for each sample. The Biofilm thickness was statistically analyzed using ANOVA.

### 2.4. Hemolysis Assay

*S. aureus* was inoculated into an NB medium with different conditions (0% NaCl, 10% NaCl, and 20% NaCl) and cultured with shaking at 140 rpm for 48 h at 37 °C. There were three replicates for each treatment. The NB liquid medium containing bacteria (1 × 10^8^ cells) was centrifuged at 12,000 rpm for 5 min at 4 °C. The supernatant was aseptically filtered (0.22-μm pores), and 500 μL of the filtered sample was poured into a centrifuge tube. Then, 25 μL of sterile defibrinated sheep blood was added, shaken gently to mix, and incubated at 37 °C for 2 h. After centrifugation at 3000 rpm for 1 min, 200 μL of the supernatant was utilized, and the absorption at 450 nm (OD_450_) was measured. A statistical analysis of the hemolysis data was performed using ANOVA.

### 2.5. Coagulase Assay

To prepare the coagulase assay, 0.5 mL of sterile normal saline was withdrawn using a pipette and added to a vial of freeze-dried plasma (rabbit plasma) and shaken until fully dissolved. Then, 0.3 mL of the stress-treated bacterial solution (0% NaCl, 10% NaCl, and 20% NaCl) was added to the dissolved rabbit plasma. After mixing evenly, it was placed in an incubator at 37 °C for static culture. It was observed once every 0.5 h for a total of 6 h. Three replicates were performed for each treatment. If coagulation occurred (a clot or coagulation with a volume exceeding one-half the total volume is positive), the plasma coagulase of the tested bacterial solution was considered to be positive. Otherwise, it was considered to be negative.

### 2.6. Transcriptome Analysis

Two samples from each treatment were selected for the transcriptome analysis. Total RNA extraction, library construction, and transcriptome sequencing were performed. Total RNA was extracted using the TRIzol reagent (Takara Bio, Otsu, Japan) according to the manufacturer’s instruction and its purity and quantity were tested using the Agilent 2100 Bioanalyzer (Agilent Technologies, Santa Clara, CA, USA). Subsequently, rRNA was removed with the Ribo-Zero™ Magnetic Kit b’j (Illumina). mRNA was fragmented into small pieces. Random hexamer primers were used for the cDNA synthesis. Finally, sequencing was carried out using the Illumina Hiseq 4000 platform (Majorbio Bio-Pharm Technology Co., Ltd., Shanghai, China) [18].

The fragments per kilobase of transcript per million fragments (FPKM) were used to calculate gene expression levels [19]. FDR ≤ 0.05 and |log_2_FC| ≥ 1 were set as the thresholds for determining significantly differentially expressed genes (DEGs). Furthermore, Gene Ontology (GO) enrichment and the Kyoto Encyclopedia of Genes and Genomes (KEGG) enrichment analysis of the DEGs were analyzed using the R package. An adjusted *p* value ≤ 0.05 was used as a threshold to determine significant enrichment. The transcriptomic data were deposited in the NCBI Sequence Read Archive under the following accession code: PRJNA764353.

### 2.7. Quantitative Real-Time PCR Experimental Validation

Quantitative real-time PCR (qRT–PCR) was adopted to detect the expression of 10 selected DEGs to validate the Illumina sequencing results. The Primer 5 program was used to design the gene-specific primers. The primer sequences are listed in Appendix A. The 20 µL reaction system contained 10 μL 2 × SYBR^®^ Premix Ex TaqTM II, 0.4 μL of each of the forward and reverse primers, 1 μL of the cDNA template, and 7.4 μL of nuclease-free water. The cycling conditions were performed as follows: pre-denaturation at 95 °C for 10 s; denaturation at 95 °C for 10 s; annealing at 55 °C for 10 s; extension at 72 °C for 20 s, in 40 cycles; and three replicates were performed for each DEG. The relative expression of DEGs was calculated according to the 2^−∆∆Ct^ method with 16S rDNA used as the housekeeping gene. A statistical analysis of the relative expression was performed using ANOVA.

### 2.8. Metabolome Analysis

Sample preparation and extraction were performed as follows: bacterial cells were fully ground in liquid nitrogen and 60% methanol was added to extract metabolites. Then, the extracts were nitrogen-dried and freeze-dried. After drying, the samples were successively mixed with methoxyamine pyridine hydrochloride (Aladdin, China), bis(trimethylsilyl)trifluoroacetamide (BSTFA, Aladdin, China), and Chlorotrimethylsilane (TMCS, Aladdin, China). Finally, heptane was added to complete derivatization. Three samples were selected from each treatment condition for GC–MS [20]. 

GC–MS and SIMCA-P + ver14 software were utilized to identify and analyze the metabolites [21]. An orthogonal partial least squares discriminant analysis (OPLS-DA) was employed to maximize the metabolome differences between the two samples. Metabolites with a variable importance in the projection (VIP) value > 1 and *p* value ≤ 0.05 were regarded as differential metabolites (DMs). MetaboAnalyst 5.0 was used to perform the KEGG analysis.

### 2.9. Correlation Analysis of Transcriptome and Metabolome

Cytoscape software was utilized to visualize the relationships between the metabolome and transcriptome based on Spearman’s correlation coefficients. A *p* value ≤ 0.05 was set as the threshold, and the network relationship diagram was drawn. Then, the hub genes were screened. In addition, the hub genes were analyzed using GO and KEGG functional annotation.

## 3. Results

### 3.1. High Salinity Stress Affects Cell Growth and Morphology

As the salinity increased, the growth activity of cells decreased gradually (Figure 1A). The inhibitory effect of high salt on cell growth was greater than that of medium salt. In this study, the bacterial cells survived in the 20% NaCl group. This demonstrated the high salt tolerance of *S. aureus* ATCC 27217.

The control group cells presented with a spherical shape and smooth surface, which is a typical feature of *S. aureus* cells (Figure 1B). After treatment with 10% NaCl, most cells showed little change in morphology (Figure 1C). After treatment with 20% NaCl, the cytomembrane of a few of the cells ruptured, and the contents of the cells were released (Figure 1D).

With an increased salinity, the growth activity of *S. aureus* was further inhibited, and the cell morphology was destroyed.

### 3.2. High Salinity Stress Affects Biofilm Formation

CLSM images of bacterial cells labeled with FITC-ConA and PI are shown in Figure 2A–C (representative schematic diagrams). The distribution of the cells and the biofilm were assessed. Green fluorescence and red fluorescence were indicative of biofilm and cell DNA, respectively. After treatment with increasing salinity, the biofilm thickness increased and then decreased. Furthermore, significant differences were identified between the adjacent groups (Figure 2D). These results suggest that an increased salt concentration promoted biofilm formation in the low salt range (<10% NaCl), while an increased salt concentration inhibited biofilm formation in the high salt range (>10% NaCl).

### 3.3. High Salinity Stress Affects Virulence

To determine whether NaCl affects the virulence of *S. aureus* ATCC 27217, we measured hemolytic activity and coagulase expression. After the salt stress intensity increased, the hemolytic activity showed a continuous downward trend, and the differences between different treatment groups were significant (Figure 3). Our results showed that high salinity stress had a significant inhibitory effect on the hemolytic activity of *S. aureus.*

The coagulase results showed that the control group underwent coagulation first, followed by the 10% NaCl group; however, the 20% NaCl group did not form a coagulation state until the sixth hour (Appendix A). Different NaCl concentrations inhibited the expression of *S. aureus* coagulase in a dose-dependent manner, thereby reducing the invasiveness of *S. aureus.*

### 3.4. Transcriptome Responses to Salinity Stress

A total of 1007, 1150 and 172 DEGs were identified in the 0% NaCl vs. 10% NaCl comparison, 0% NaCl vs. 20% NaCl comparison, and 10% NaCl vs. 20% NaCl comparison, respectively, (Figure 4). Compared with the 0% NaCl group, both the 10% NaCl group and the 20% NaCl group mobilized a large number of DEGs to respond to salinity stress. Moreover, the number of DEGs in the 0% NaCl vs. 20% NaCl comparison was higher than that in the 0% NaCl vs. 10% NaCl comparison. It demonstrated that *S. aureus* requires the mobilization of more DEGs to adapt and survive under higher salt stress.

The following four change patterns were observed after the salt concentration gradually increased (Figure 5): increase/increase, increase/decrease, decrease/decrease, and decrease/increase. The results showed that 6, 4, 72, and 32 DEGs were identified in each pattern. Detailed annotation information is listed in Appendix A. Therefore, we believe these DEGs are candidate target genes in the salt stress response. 

In total, nine predicted KEGG pathways were enriched significantly (Appendix A). They include *S. aureus* infection, aminoacyl-tRNA biosynthesis, ribosome, fatty acid degradation, monobactam biosynthesis, butanoate metabolism, benzoate degradation, tyrosine metabolism, purine metabolism, glycine, serine, and threonine metabolism. These pathways were mainly involved in amino acid metabolism, translation, infectious diseases, purine metabolism, carbohydrate metabolism, and lipid metabolism.

### 3.5. qRT–PCR Validation

The mRNA transcription levels of ten DEGs were quantified to further verify the accuracy of RNA sequencing. Figure 6 shows that the expression levels of the selected DEGs were highly consistent with those suggested by the RNA-Seq results. Therefore, the qRT–PCR results confirmed the reliability and accuracy of the transcriptome data.

### 3.6. Metabolomic Responses to High Salinity Stress

In this analysis, a total of 75 metabolites were identified under high salinity stress (Appendix A). The differences between samples were assessed by performing a principal component analysis (PCA). As seen in Figure 7A, endogenous metabolites of the three treatment groups were located in three different quadrants, implying that there were obvious differences among the samples. In total, 37 DMs were screened after OPLS-DA (Appendix A). The DMs were subjected to a KEGG pathway analysis to identify important contributions in response to high salt stress. The metabolic pathways in the different groups are displayed in Figure 7B–D. These pathways mainly involve inositol phosphate metabolism; glycine, serine, and threonine metabolism; glyoxylate and dicarboxylate metabolism; alanine, aspartate, and glutamate metabolism; glutathione metabolism; sulfur metabolism; the TCA cycle; glycolysis/gluconeogenesis; and pyruvate metabolism (Appendix A). Therefore, the metabolic results suggest that signal response, amino acid regulation, and carbohydrate metabolism regulation are the main pathways by which *S. aureus* ATCC 27217 adapts to high salinity stress.

### 3.7. Integrated Analysis of Transcriptome and Metabolomic Data

As mentioned above, 53 key candidate genes and 37 DMs were identified. Then, transcriptome and metabolomic data were combined to obtain a Spearman correlation network of *S. aureus* ATCC 27217 in conditions causing high salinity stress. As a result, 90 nodes were connected in the network with 654 edges in the image obtained with Cytoscape. By analyzing the edges between more than 15 nodes, 20 hub genes (ABC.CD. P, adhP, arlR, atl, clpL, crtO, E2.3.1., fib, K07078, lctP, LDH, lysC, mtsA, mtsB, mtsC, ohyA, opuD, pflA, pstS, and spoVG) responding to salt stress were identified (Figure 8A). Moreover, these genes were all observed to be in a continuous downward adjustment mode. In total, 19 hub genes were assigned to the following three functional categories: 49 biological processes (BP), 14 cellular components (CC) and 23 molecular functions (MF) (Figure 8B). Most hub genes were categorized into a single-organism process, catalytic activity, cellular process, and metabolic process. In addition, 11 hub genes were assigned to 30 KEGG pathways (Figure 8C). These pathways mainly involve the transfer system (ABC transporters, two-component system) and carbon and nitrogen metabolism (glycolysis/gluconeogenesis, fatty acid degradation, pyruvate metabolism, tyrosine metabolism, cysteine, and methionine metabolism, lysine biosynthesis, glycine, serine, and threonine metabolism). Overall, *S. aureus* showed adaptive changes after undergoing high salinity stress. These hub genes require in-depth functional verification and provide the basis for further research.

## 4. Discussion

*S. aureus* can sense exogenous salt-stimulated signaling and then activate the stress response, changing its metabolic and physiological states, thereby improving its ability to survive [22]. In this study, phenotypic identification was combined with transcriptomic and metabolomic analyses to comprehensively elucidate this mechanism. We found that biofilm formation, virulence, transfer system, and osmotic regulation were the main changes that occurred in *S. aureus* ATCC 27217 under high salinity stress.

### 4.1. Biofilm Formation

EPS is the main component of biofilm and contributes to the stability of cells [23]. Biofilms are recalcitrant to harsh environments and can protect bacteria from severe conditions by acting as “protective clothing” [24]. Autolysin IsaA modulates cell wall homeostasis as a protein implicated in biofilm formation [25]. In the increase/decrease change pattern, we found that *IsaA* was a key candidate target gene in response to salt stress. From the characterization results in our study, we observed that the amount of biofilm formation increased and then decreased, which is consistent with the change in the expression of *IsaA*. Lopes et al. [26] revealed that the deletion of *IsaA* led to a significant decrease in biofilm formation. Therefore, we speculate that when *S. aureus* ATCC 27217 is subjected to salt stress, *IsaA* positively regulates and participates in biofilm formation. 

### 4.2. Virulence

As an exotoxin, hemolysin is an important virulence factor that causes *S. aureus* infection, and it is also the leading cause of red blood cell hemolysis [27]. Plasma coagulase is an aggressive enzyme whose role is to deposit and coagulate fibrin in the plasma on the surface of bacteria to prevent phagocytosis by phagocytes [28]. The hemolysin and coagulase produced by *S. aureus* can cause sepsis and purulent infections in humans and, in severe cases, may be fatal [29]. This study found that high salinity stress inhibited the hemolytic activity and expression of coagulase in *S. aureus* ATCC 27217; that is, high salinity stress inhibited the expression of virulence factors, and this inhibition occurred in a dose-dependent manner.

Staphylococcal enterotoxins represent a group of toxin proteins and the most common cause of staphylococcal food poisoning [30]. They are the main factors that contribute to vomiting and diarrhea. Staphylococcal enterotoxins have strong heat resistance, so they are difficult to remove by heating [31]. In the RNA-seq results, we found that the staphylococcal enterotoxins gene *set* was significantly down-regulated by 2.1-fold (0% NaCl vs. 20% NaCl). It can be seen that an increase in salinity reduces the expression of *set*. Sihto et al. [32] studied the expression of staphylococcal enterotoxin genes under NaCl stress at the mRAN level, and demonstrated that NaCl stress led to a decrease in the expression of staphylococcal enterotoxin genes in *S. aureus*. This is consistent with the result we obtained in our study. It can be seen that under high salt stress, the down-regulation of the enterotoxin gene represents a decline in its virulence.

*Atl* plays a key role in bacterial adhesion, internalization, and biofilm formation [33]. Porayath et al. [34] emphasized that atl is one of the vital proteins in staphylococci. Atl assists staphylococci in binding to many kinds of host cell components during colonization and infection. Efb, a fibrinogen-binding protein secreted by *S. aureus*, is an important virulence factor in staphylococcal infection [35]. Efb attracts fibrinogen to the surface of the bacteria, thereby providing *S. aureus* with the ability to cause both superficial and deep infections [36,37]. In this study, the sequential downregulation of the hub genes *atl* and *efb* indicated the reduced ability of bacteria to colonize and infect. In the significant metabolic pathway, the number of downregulated genes (twenty-five) mapped to *S. aureus* infection was greater than the number of upregulated genes (four); therefore, we inferred that this metabolic pathway was downregulated. Masters et al. [38] demonstrated that a large number of virulence factors and toxins are produced by *S. aureus* during infection. Therefore, we suggest that the downregulation of *S. aureus* infection pathway is also one of the reasons for the downregulation of virulence. 

When microorganisms have to engage in strong metabolic alterations for survival under severe stress, they downregulate the effort directed to cellular functions which are not of core significance for survival, such as virulence [39]. Herein, we believe that virulence was downregulated to reduce the depletion of nonessential cellular functions to survive in a highly salt-stressed environment.

### 4.3. Transfer System

DM myo-inositol (VIP = 1.27395) was downregulated significantly (*p* < 0.01) in 0% NaCl vs. 10% NaCl. Simultaneously, the impact score (impact score = 0.625) of the inositol phosphate metabolism was the highest in this group. The transformation of various phosphoinositide molecules occurs in this pathway. Myo-inositol is the only base molecule involved in the conversion of all phosphoinositide molecules, so a decrease in inositol content leads to the downregulation of this pathway (Figure 9). Previously, researchers reported that phosphoinositides participate in various life-related activities, such as stress resistance, DNA repair, signal response, and RNA transport [40,41,42]. These molecules play various functions in different metabolic pathways. Taken together, we speculate that *S. aureus* ATCC 27217 reduces the conversion rate of functional molecules by downregulating inositol phosphate metabolism, thereby reducing the unnecessary activation of molecular functions to alleviate the survival crisis caused by high salt stress.

MtsABC is a typical ABC transporter system composed of lipoprotein mtsA, ATP-binding protein mtsB, and hydrophobic integral membrane protein mtsC [43]. In this study, the expression of the hub genes *mtsA*, *mtsB*, and *mtsC* was continuously downregulated as the concentration of salt stress increased (Figure 9). ABC transporters regulate the homeostasis of ions in bacteria, and participate in nutrient absorption, signal molecule output, and the elimination of toxic compounds. They play an essential role in cellular defense [44]. Thus, this finding suggests that the MtsABC transport system maintains intracellular homeostasis in *S. aureus* ATCC 27217 by reducing ion transport and signaling, thereby protecting cells from high salt damage.

In *S. aureus* ATCC 27217, 17 DEGs were associated with the aminoacyl-tRNA biosynthesis pathway, and 16 DEGs were upregulated. Therefore, the aminoacyl-tRNA biosynthesis pathway showed an overall upward activation trend. This pathway can transport amino acids to promote protein synthesis. It ensures that gene sequences can be accurately transcribed and translated into proteins, thereby maintaining cellular homeostasis [45]. From the pathway analysis results, we also found that salt stress leads to abnormal amino acid metabolism. Hence, *S. aureus* ATCC 27217 improved cellular homeostasis by upregulating aminoacyl-tRNA biosynthesis, ensuring the rigor and diversity of its life forms and resisting salt stress.

### 4.4. Osmotic Adjustment

The accumulation of compatible substances and osmotic adaptation are critical processes by which cells regulate high osmotic pressure [46,47]. In glycine, serine and threonine metabolism, two DEGs (*betA* and *betB*) were mapped to the betaine biosynthesis pathway. Additionally, both DEGs were upregulated, so we concluded that the upregulation of the betaine biosynthesis pathway was activated (Figure 9). Ming et al. [16] demonstrated that the upregulation of betA and betB in *S. aureus* regulates betaine synthesis and transformation, thereby increasing osmotic stability. Brown et al. [48] suggested that this osmoregulation is capable of protecting the active structure of proteins and nucleic acids. It was also previously reported in foodborne bacteria such as *Vibrio parahaemolyticus* [49], and *Listeria monocytogenes* [50]. All of these results indicate the upregulation of the betaine biosynthesis pathway can regulate osmotic pressure and protect *S. aureus* ATCC 27217 intracellular biomacromolecules against salt stress.

Previous studies reported that after undergoing low-concentration salt stress, *S. aureus* accumulates a great amount of L-proline for osmotic regulation [51,52]. This mechanism was also reported in foodborne bacteria such as *Staphylococcus saprophyticus* [53], *Vibrio parahaemolyticus* [48] and *Listeria monocytogenes* [54]. In the present study, the content of DM L-proline was significantly increased in the environment of 10% NaCl, which was consistent with previous studies. However, after the bacteria were transferred to the 20% NaCl environment, the L-proline content decreased. We speculate that the decrease in the L-proline content may be due to the high salt (20%NaCl) damage caused to the cell structure, resulting in the release of proline and further contents outside the cell. Eventually, the normal metabolism of proline becomes disturbed. The results of this study provide a scientific basis for the further exploration of the changes in the metabolic system of *S. aureus* in a high osmotic environment.

In the present study, although we employed an omics-combined characterization approach, the mechanism of *S. aureus* survival in salted foods was investigated from a genetic and metabolic perspective. However, the findings were focused on *S. aureus* ATCC 27217, and it is unclear whether they are applicable to other *S. aureus*. In addition, two biological replicates were performed for the transcriptome and three biological replicates for the metabolome. Nevertheless, there is still a deficiency in the number of replicates performed for the samples.

## 5. Conclusions

Through a comprehensive analysis of the transcriptome and metabolome, we provided valuable information about the regulatory network of *S. aureus* ATCC 27217 in response to high salinity stress. Virulence metabolism and signaling pathways were downregulated to reduce the consumption of non-essential cellular functions. The synthesis pathway of osmotic protective substances was upregulated to protect the stability of intracellular life substances and osmotic pressure. The aminoacyl-tRNA biosynthesis pathway was upregulated to maintain the rigor and diversity of cellular life forms. The regulation of these metabolic pathways was an essential method by which *S. aureus* was able to self-protect against salinity stress. They could serve as potential targets to help prevent bacterial infections. The results of this work will provide additional knowledge on the biological strategies employed by salt-tolerant bacteria in salted foods.

## Figures and Tables

**Figure 1 foods-11-01503-f001:**
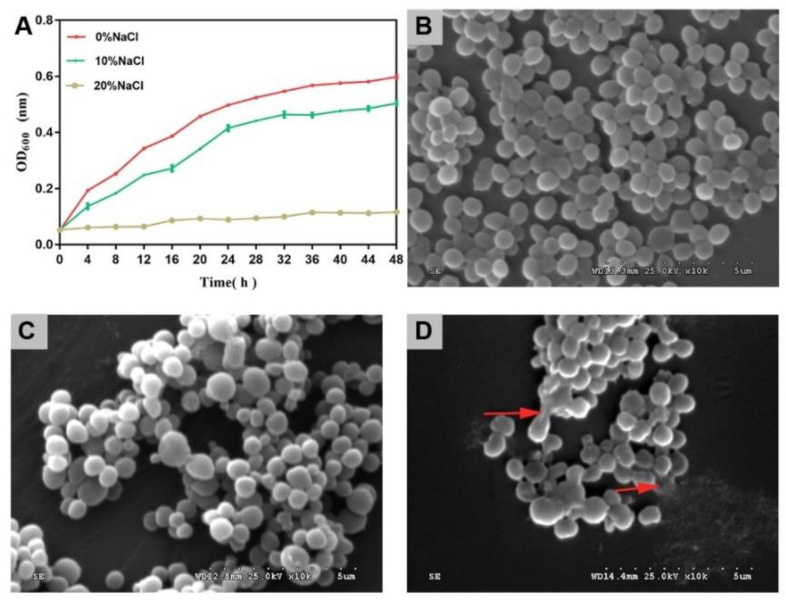
Cell growth and SEM images of *S. aureus* during salinity changes. (**A**) Growth curves. The error bars indicate the standard deviation (N *=* 3). (**B**) SEM image of *S. aureus* in control group. (**C**) SEM image of *S. aureus* in 10% NaCl group. (**D**) SEM image of *S. aureus* in 20% NaCl group. ×10K-fold.

**Figure 2 foods-11-01503-f002:**
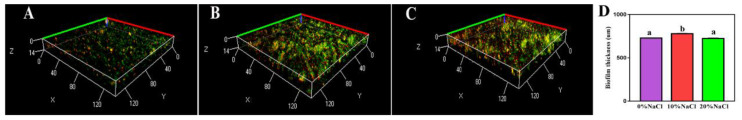
CLSM images of treated *S. aureus* biofilm cells. (**A**) Control group; (**B**) 10% NaCl group; (**C**) 20% NaCl group; (**D**) thickness of biofilm.

**Figure 3 foods-11-01503-f003:**
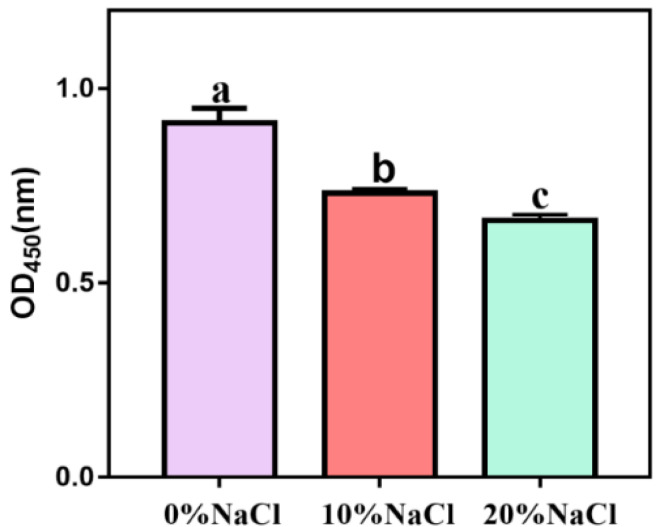
Hemolytic activity of *S. aureus* during change in salinity.

**Figure 4 foods-11-01503-f004:**
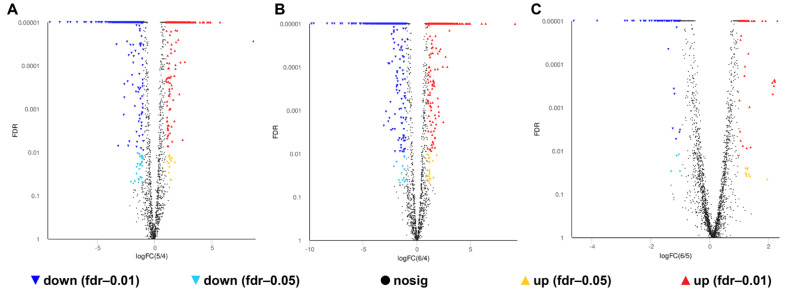
Volcano plot of genes. The red and green data points indicate upregulated and downregulated DEGs, respectively. (**A**): 0% NaCl vs. 10% NaCl; (**B**): 0% NaCl vs. 20% NaCl; (**C**): 10% NaCl vs. 20% NaCl.

**Figure 5 foods-11-01503-f005:**
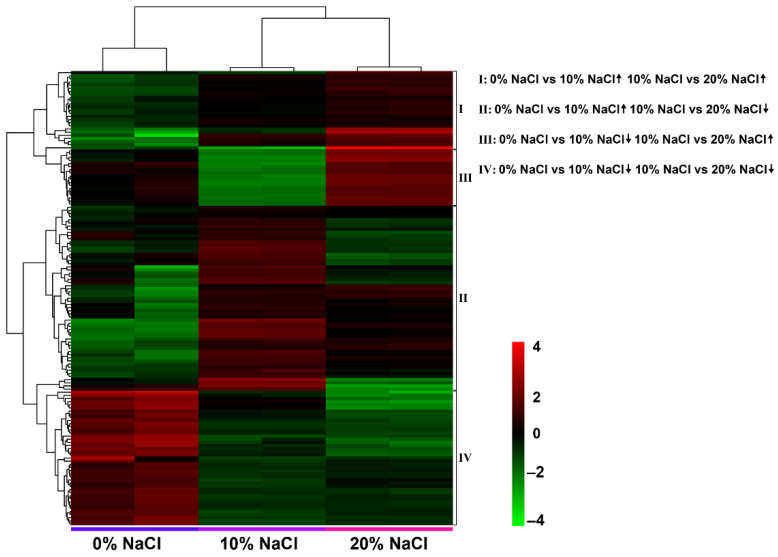
DEGs related to salt stress identified with pairwise comparison of four expression patterns. Heatmap of the four expression patterns of DEGs after a gradual increase in salt concentration.

**Figure 6 foods-11-01503-f006:**
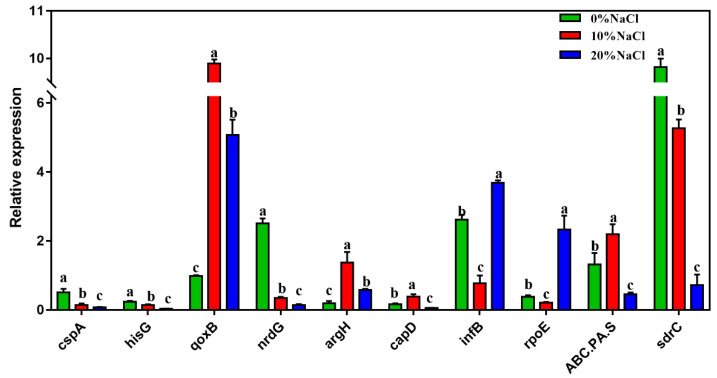
The relative expression of the validation genes. The *Y*-axis represents the expression of selected genes. Different letters correspond to statistically significant differences (*p* < 0.05) between groups.

**Figure 7 foods-11-01503-f007:**
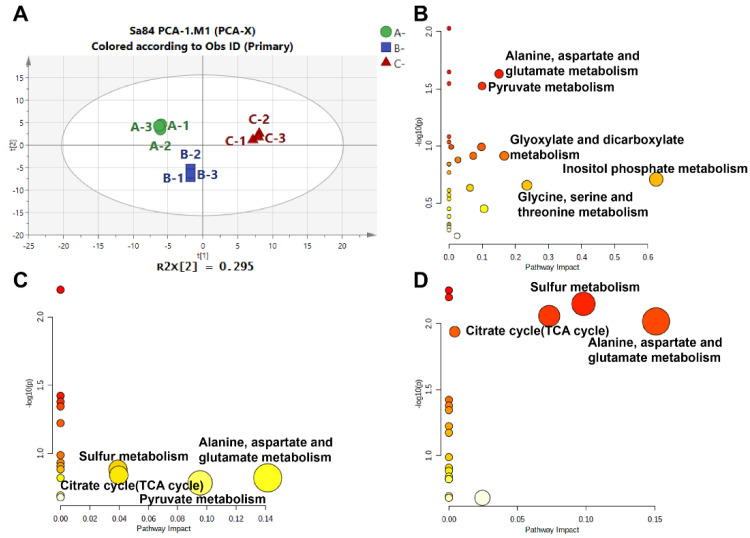
Overview of the metabolome analysis results. (**A**) PCA score plot of the overall metabolites expression; (**B**) KEGG pathways in 0% NaCl vs. 10% NaCl comparison; (**C**) KEGG pathways in 10% NaCl vs. 20% NaCl comparison; (**D**) KEGG pathways in 0% NaCl vs. 20% NaCl comparison.

**Figure 8 foods-11-01503-f008:**
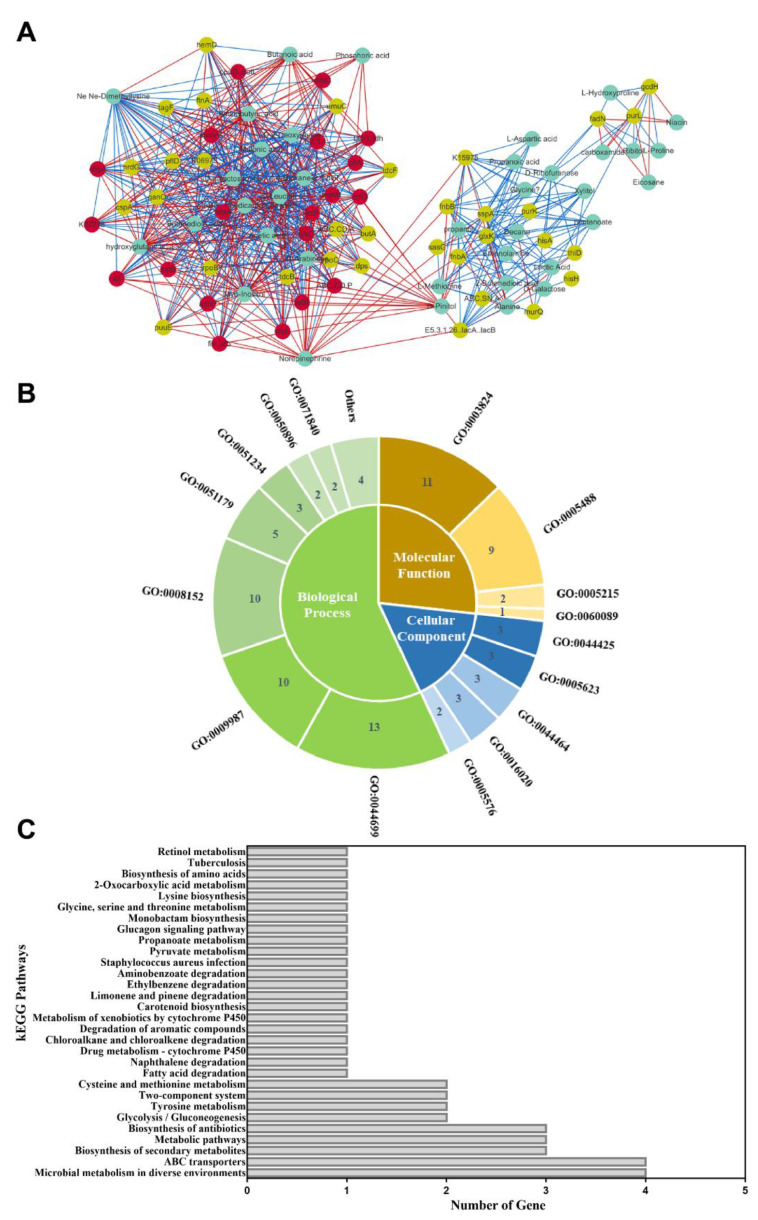
The results of the integration analysis. (**A**) Relational Graph of key candidate genes and DMs. Green node represents DM, yellow node represents candidate gene, and red node represents hub gene. The red and green lines represent positive and negative correlations, respectively. The thicker the line, the larger the correlation coefficient, and the thinner the line, the smaller the correlation coefficient; (**B**) GO functional classification; Biological processes, molecular functions, and cellular components associated with hub genes; (**C**) Pathway analysis of hub genes. The *X*-axis represents the number of annotated genes. The *Y*-axis represents the pathway terms.

**Figure 9 foods-11-01503-f009:**
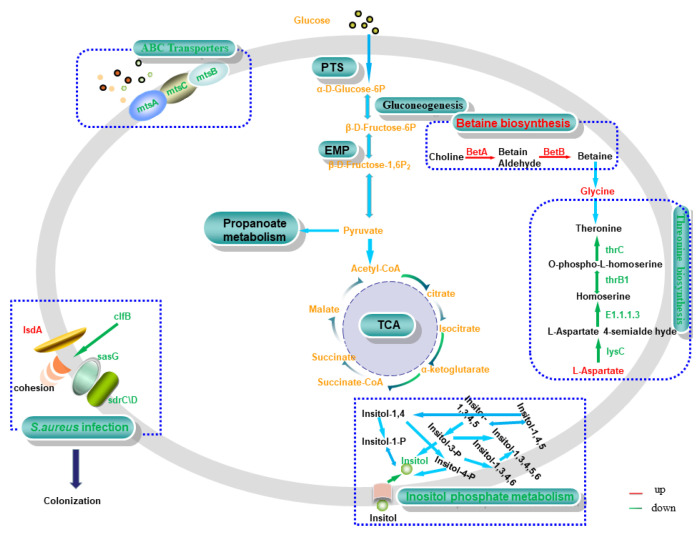
Relevant pathways in *S. aureus*. Red and green represent upregulated and downregulated expression, respectively. Matched genes include choline dehydrogenase (*betA*), betaine-aldehyde dehydrogenase (*betB*), manganese ABC transporter substrate-binding protein (*mtsA*), membrane protein ABC transporter permease (*mtsB*), phosphonate ABC transporter ATP-binding protein (*mtsC*), clumping factor B (*clfB*), heme transporter IsdA (*isdA*), Ser-Asp rich fibrinogen-binding protein (*sdrC/D*), surface protein G (*sasG*), aspartate kinase (*lysC*), homoserine dehydrogenase (*E1.1.1.3*), homoserine kinase (*thrB_1_*), and threonine synthase (*thrC*).

## Data Availability

The data presented in this study are openly available in the NCBI Sequence Read Archive under accession code PRJNA764353.

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
