# Peer review of "The Response and Survival Mechanisms of Staphylococcus aureus under High Salinity Stress in Salted Foods"

_foods, 2022, doi:10.3390/foods11101503_

Round 1

Reviewer 1 Report

In this manuscript, the authors exposes the regulation mechanisms and genes expression levels to Staphylococcal bacteria survival in high salinity environment. For this, phenotypic, transcriptomic and metabolomic analyses were performed, and this represent a lot of work. So, first, I would like thank the authors for this contribution. The results tend to show that in high salinity conditions, the non-essential functions are downregulated, this would in order to preserve stability. This work is very interesting because a diversity of analysis was performed to conclude this.

However, I have some commentaries about the methods essentially.

In the first part concerning the Strain and culture conditions, the authors said N=3, in the line 74. I didn’t understand if it means that for each conditions, three replicates were performed or if it refers to the three conditions. When I looked the results, I had the impression there is not replicates. So I think it would be better to reformulate and explains this in the method part.

Concerning the hemolytic analyses, I didn’t understand where does it come from the liquid containing bacteria. The authors could be precised the manipulation to obtain it, this comment is also available to biofilm and coagulase analyses.

In the result part, and more precisely the biofilm part, I would like to have more explanation about the Z Stack calculation. And in the figure 2D, I would like know how differences in histogram were obtained. I think the authors should explain the statistics used and the replicate number to obtain a statistically supported result. This comment is also available for the figure 3 because I didn’t understand how the authors obtained the standard deviations to represented them on histograms.

About the transcriptomic analyses, It seems that 2 analyses were performed for each conditions, as represented on the figure 5, is it this ? If yes, I think it should be precise on the method section to improve the comprehension of manuscript.

About this part and RT-PCR, I think it is very important to have validated results using the two methods, and this allow to support conclusion of transcriptomic analyses. It is for this, I think that the authors should precise PCR conditions used in method section. And they should precise again the number of realized replicates and the statistics used to compare the results.

About the figure 8, I observed a mistake in the legend, nodes are in green and lines in blue. I was wondering If there is a threshold about the correlation representation, and if yes, I think it would be interesting to indicate this.

About the discussion, I think the authors should be more nuanced in their conclusions because analyses were performed on only one strain whereas Staphylococcal aureus is a bacterial species highly diverse. And I have a question for the authors: Have these mechanisms also been observed in other bacteria in food, as Bacillus or Salmonella ?

Author Response

请参阅附件

Reviewer 2 Report

Major changes

-Water Activity is not discussed in introduction, methods or discussion at all, this is the unit of used to describe water stress and should be added.

-Methods have major gaps that must be addressed-work is unrepeatable by others in current form due to poor method description. Culture medium is not described! Nor is water activity.

-“as previously described” is used way to much, method needs to be summarized so reader understands it and can repeat it! Currently this cannot be done.

-full name and suppliers of FITC-ConA and PI not present. Again work cannot be repeated.

-figure 3 has error bars...when sample size =3! This is very inappropriate and implies a statistical relationship when none should be.

Discussion

-Toxin production ramifications not discussed? This is why S. aureus a problem, what happens to its emetic and diarrheal toxin expression? Emetic toxin also survives cooking-so it is critical to virulence.

-limitations not discussed, there are several major ones (like 1 strain was studied, sample size = 3 etc, these limitations effect what conclusions can be drawn and identify areas of future work)

Minor

-in biofilm analysis(84-93) replace ‘can combine’ with ‘bind’, much simpler and clearer

-line 161 you didn’t have <10% conditions, you just had at 20%

-S. aureus not in italics for half of manuscript.

-line 206 “infectious disease”, meaning not clear, do you mean virulence factors or genes linked to pathogenicity or toxin producing genes?? Needs to be clarified

-line 263-264 good strong statement, needs a reference!

-line 281-308, this is a very large intense technical paragraph, recommend breaking up into small paragraphs to increase clarity.

Reviewer 3 Report

This research makes a good contribution to the study of resistance of S. aureus regarding food contamination.

The paper is well structured, materials and methods are clearly and in detail described and the results obtained follow the discussion.

Authors use modern methodology with the application of traditional microbiological methods that complement each other excellently in order to obtain reliable results.

Obtained results represent an excellent basis for further research in this direction and therefore I suggest this paper to be accepted in its original form.

Minor suggestions:

line 72-in which medium broth was S. aureus incubated?

line 73-74-which medium?

line 84- how was bacterial volume adjusted?

line 95-which liquid? describe in more detail. What was the number of cells? The speed of centrifuge should be written uniformly, either x g or in rpm

line 102-are bacterial cells in sterile saline already added? In which number?

line 104-describe briefly method by Roberson.

Round 2

Reviewer 2 Report

Authors have responded to reviewers comments well and addressed weaknesses in paper that were identified.